# Seesaw conformations of Npl4 in the human p97 complex and the inhibitory mechanism of a disulfiram derivative

Man Pan [1,4], Qingyun Zheng[2,4], Yuanyuan Yu[1,4], Huasong Ai[2], Yuan Xie [1], Xin Zeng[3], Chu Wang [3], Lei Liu [2✉] & Minglei Zhao [1✉]

p97, also known as valosin-containing protein (VCP) or Cdc48, plays a central role in cellular protein homeostasis. Human p97 mutations are associated with several neurodegenerative diseases. Targeting p97 and its cofactors is a strategy for cancer drug development. Despite significant structural insights into the fungal homolog Cdc48, little is known about how human p97 interacts with its cofactors. Recently, the anti-alcohol abuse drug disulfiram was found to target cancer through Npl4, a cofactor of p97, but the molecular mechanism remains elusive. Here, using single-particle cryo-electron microscopy (cryo-EM), we uncovered three Npl4 conformational states in complex with human p97 before ATP hydrolysis. The motion of Npl4 results from its zinc finger motifs interacting with the N domain of p97, which is essential for the unfolding activity of p97. In vitro and cell-based assays showed that the disulfiram derivative bis-(diethyldithiocarbamate)-copper (CuET) can bypass the copper transporter system and inhibit the function of p97 in the cytoplasm by releasing cupric ions under oxidative conditions, which disrupt the zinc finger motifs of Npl4, locking the essential conformational switch of the complex.

[1] Department of Biochemistry and Molecular Biology, The University of Chicago, Chicago, IL 60637, USA. [2] Tsinghua-Peking Center for Life Sciences, Department of Chemistry, Tsinghua University, 100084 Beijing, China. [3] Peking-Tsinghua Center for Life Sciences, College of Chemistry and Molecular Engineering, Peking University, 100871 Beijing, China. [4] These authors contributed equally: Man Pan, Qingyun Zheng, Yuanyuan Yu. ✉email: lliu@mail.tsinghua.edu.cn; mlzhao@uchicago.edu

P97 (also known as valosin-containing protein (VCP) or Cdc48 in yeast)[1] is a highly abundant cytoplasmic protein in eukaryotic cells[2]. It belongs to the protein family known as ATPase associated with diverse cellular activities (AAA + protein superfamily)[3]. p97 has two tandem ATPase domains named D1 and D2 and an additional N domain at the N-terminus (Fig. 1a). Similar to many AAA + proteins, p97 converts chemical energy from ATP hydrolysis to mechanical forces, which then relocate or unfold ubiquitinated substrates[4]. In vivo, p97 functions as a homomeric hexamer, a 540 kDa molecular machine with twelve copies of ATPase domains organized into two rings[5]. p97 plays a central role in cellular proteostasis. More than 30 mutations of human p97 have been discovered, and these are associated with a number of neurodegenerative diseases, including inclusion body myopathy, frontotemporal dementia, and familial amyotrophic lateral sclerosis[6,7]. Targeting p97 to disrupt cellular proteostasis is also a strategy for cancer therapy[8].

One unique feature of p97 is that it can interact with a remarkable diversity of cofactors to participate in a variety of cellular pathways[9]. Most of these cofactors engage ubiquitinated substrates. The most versatile cofactor is a heterodimer consisting of two proteins: Npl4 and Ufd1 (denoted Npl4/Ufd1), which are likely involved in more than half of the cellular processes in which p97 participates[2]. A well-studied example is endoplasmic reticulum-associated degradation (ERAD), in which Cdc48, the yeast homolog of p97, recruits Npl4/Ufd1 to extract poly-ubiquitinated misfolded membrane proteins from the ER membrane, then processes and delivers the extracted proteins to the proteasome for degradation[10,11]. In particular, the FDA-approved drug disulfiram (tetraethylthiuram disulfide, trade name Antabuse), which has been used for a long time in the treatment of alcoholism, has recently been discovered to be effective against various cancer types in preclinical studies by targeting the cofactor Npl4[12].

Recently, the cryo-electron microscopy (cryo-EM) structures of the thermophilic fungus Cdc48-Npl4/Ufd1 complex[13] and the yeast Cdc48-Npl4/Ufd1-Eos complex[14] were reported. In both structures, Npl4 interacts with one of the N domains of Cdc48

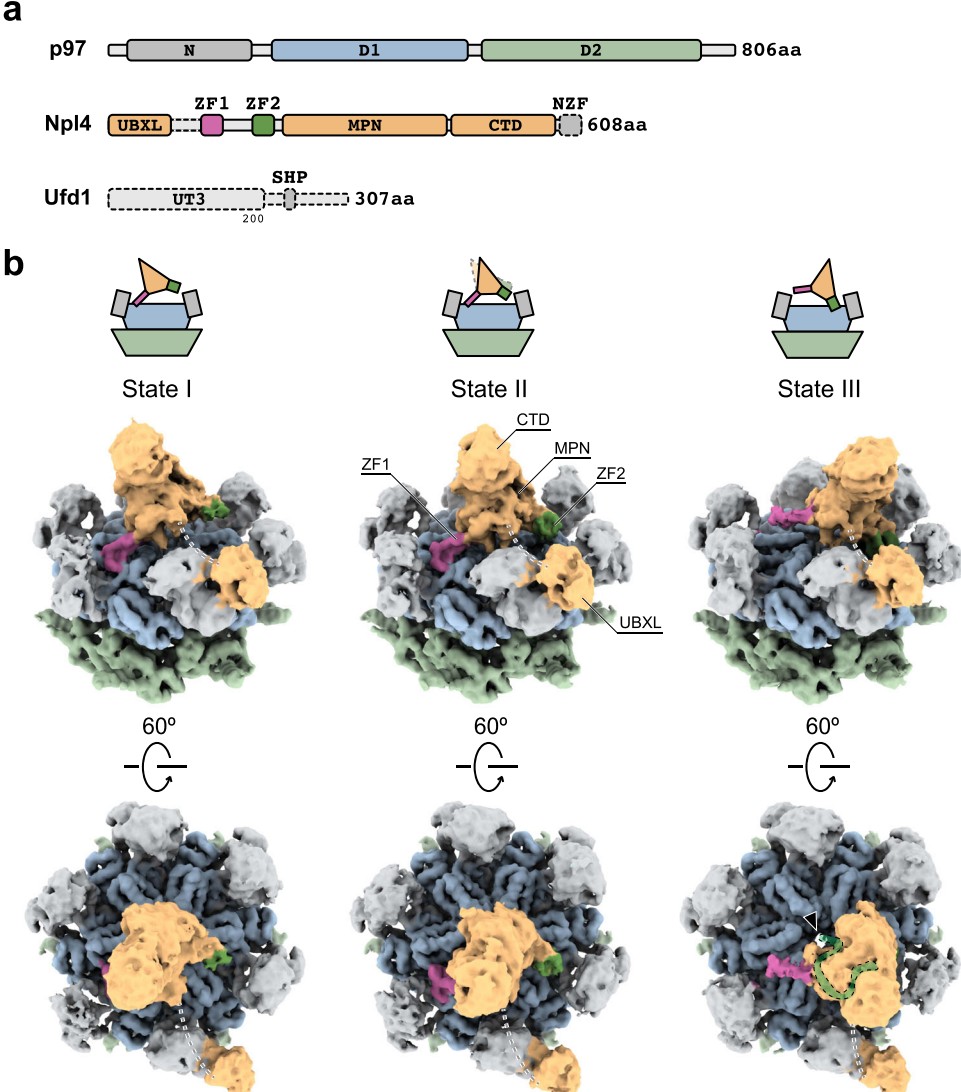

**Fig. 1 Three conformational states of human p97 in complex with Npl4/Ufd1. a** Domain architecture of human p97, Npl4, and Ufd1. Unresolved parts are shown with dotted lines. **b** Single-particle cryo-EM maps (unsharpened, 0.014 threshold) of human p97 in complex with Npl4/Ufd1. Three conformational states were resolved. The position of Npl4 relative to p97 is illustrated in the cartoons. Unresolved linkers between UBXL and ZF1 are represented by dotted lines. The central pore of the D1 ring in State III is highlighted by an arrowhead. The conserved groove of Npl4 that was found to bind an unfolded ubiquitin in the yeast complex structure[14] was marked by a green band in State III.

through its ubiquitin regulatory X-like (UBXL) domain and anchors the Mpr1/Pad1 N-terminal (MPN) domain on top of the D1 ring using both zinc finger motifs. One surprising finding was that the yeast Cdc48-Npl4/Ufd1 complex initiates substrate processing by unfolding one ubiquitin molecule in the absence of ATP binding or hydrolysis. The unfolded ubiquitin molecule binds to the groove of Npl4 and projects its N-terminal segment through both the D1 and D2 rings. The structure provided unprecedented insights into the substrate processing of Cdc48; however, it also raised a new question of how ubiquitin is unfolded and inserted into the central pore of the Cdc48 complex. More information about conformational dynamics is essential for a better understanding of this process.

Here, we investigated how human p97 interacts with its cofactor, Npl4/Ufd1, in the presence or absence of ubiquitinated substrates and ATPγS using cryo-EM and single-particle analyses. Our results showed that unlike the yeast homolog Cdc48, human p97 interacts with the cofactor Npl4/Ufd1 and its substrate in multiple conformational states without ATP hydrolysis. Furthermore, we revealed that the disulfiram derivative bis-(diethyldithiocarbamate)-copper (CuET) inhibits the unfolding activity of p97 and locks the conformational changes between p97 and Npl4/Ufd1 by releasing cupric ions under oxidative conditions. Our results provided mechanistic insights into the broad anticancer activity of disulfiram when used in combination with copper compounds.

## Results

### Three conformational states of human p97 in complex with Npl4/Ufd1.

Our work started with the structure determination of p97 in complex with the cofactor Npl4/Ufd1 to reveal the difference between the yeast and human systems. We determined the complex structure in the presence of ATPγS using cryo-EM and single-particle analyses (Supplementary Fig. 1). Using 3D classification, we resolved three different conformational states of the complex. In contrast to the complex structure of fungal Cdc48, in which Npl4 binds to the D1 ring using both zinc finger motifs (Supplementary Fig. 2c), human Npl4 binds to the D1 ring of p97 with only one zinc finger motif (Fig. 1b and Supplementary Fig. 1). In States I and II, Npl4 binds to the D1 ring of p97 using zinc finger motif 1 (ZF1, purple), whereas in State III, Npl4 binds to the same hydrophobic groove of the D1 ring using zinc finger motif 2 (ZF2, green). In both cases, the zinc finger motifs form hydrogen bonds with the backbone of an existing β-strand (F265-I269) in the large subunit of the D1 domain (Supplementary Fig. 2a, b). The difference between States I and II is that the main structure of Npl4, namely, the MPN and carboxyl-terminal domain (CTD), is rotated by ~8 ° and is displaced by ~13 Å with ZF1 as the fulcrum. Compared with States I and II, the main structure of Npl4 in State III is displayed by ~49 Å and ~38 Å, respectively (Supplementary Fig. 2d). Notably, when focused on the conserved Npl4 groove, we observed that only the center hole of p97 in State III is aligned with the groove and exposed in the top view (Fig. 1b); this was not observed in the complex structure of yeast Cdc48. Together, the conformational changes of these three states present a seesaw-like motion of Npl4 on top of the D1 ring. In all three states, the other zinc finger motif that is not interacting with the D1 ring is close to one of the N domains of p97, suggesting potential interactions that are observed in the complex structures with a polyubiquitinated substrate (Fig. 2). In addition to ZF1 and ZF2, a third zinc finger motif (NZF), which does not exist in the fungal homolog, is located at the very C-terminus of Npl4 (Fig. 1a). We were not able to resolve NZF due to the degradation of the local resolution (Supplementary Fig. 1c). A homology model of Npl4 was built

based on the crystal structure of the thermophilic fungal homologue[13]. p97 adopts a six-fold symmetrical conformation in all three states. In fact, when applying a mask around the D1 and D2 rings of p97 and imposing a C6 point-group symmetry, the resolution of the 3D reconstruction improved to 2.8 Å, which allowed de novo model building (Supplementary Figs. 1d, 2e, f, and Supplementary Table 2). A comparison between our model and the published human p97 structures suggested a very similar conformation to one of the ATPγS bound states (PDB accession code: 5ftn[15]) with an overall root mean square deviation (rmsd) of 1.095 Å. Only a minimal part of Ufd1 in complex with Npl4 was visible in the unsharpened maps, similar to the fungi complexes reported before[13,14].

### Seesaw motion of Npl4 is important for the unfolding activity of human p97.

To further investigate the role of the seesaw motion of Npl4, we determined the structure of human p97 in complex with Npl4/Ufd1 and a K48-linked polyubiquitinated substrate Ub-Eos, which has two tandem ubiquitin molecules fused to the N-terminus of the fluorescent protein mEos3.2, using single-particle cryo-EM (Supplementary Fig. 3). No additional nucleotides were supplied during the purification and vitrification in order to resolve the native conformations. Moreover, two mutations of p97, A232E, and E578Q, were introduced to stabilize the complex. It has been reported that A232E, a mutation found in patients with neurodegenerative diseases, can fix the N domain of p97 in "up" conformation and has increased affinity to Npl4/Ufd1[16,17]. E578Q, a mutation in the Walker B motif of the D2 domain, inhibits ATP hydrolysis in the D2 ring. Again, three major conformational states of the p97-Npl4/Ufd1 complex with substrate engagement were resolved (Fig. 2). In substrate-engaged States I and II, the conformations are essentially the same as those observed from States I and II without the substrate. The only change was the observation of density corresponding to a single ubiquitin on the very top of Npl4. In substrate-engaged State III, in addition to the ubiquitin density, we also found a displacement of one N domain that was "dragged" by the raised ZF1 motif, which is approximately 13 Å (Fig. 2). The density corresponding to the polyubiquitinated Ub-Eos is only visible at a lower threshold and is not resolvable (Supplementary Fig. 4a). Recently, the cryo-EM structure of yeast Cdc48 in complex with Npl4/Ufd1 and K48-linked polyubiquitinated Eos showed that an unfolded ubiquitin moiety binds in the groove of Npl4 and extends all the way into the pore of Cdc48 in the absence of ATP hydrolysis[14]. By contrast, none of the states showed any density in the groove of Npl4 (Fig. 2), suggesting that the resolved structures are prior to ubiquitin unfolding and the polyubiquitinated substrate translocating. Together, the main conformational change caused by the binding to the ubiquitinated substrate is the clear interaction between ZF1 and the N domain of p97 in State III. (Fig. 2 and Supplementary Fig. 4b). Since the resolutions of the maps do not allow model building at the atomic level (Supplementary Fig. 3d), we docked the crystal structure of the N domain[18] and the homology model of Npl4 into the density as rigid bodies, which showed that R113 in the N domain is likely involved in the interaction (Supplementary Fig. 4b). To test if the interaction is relevant for the function of p97, we mutated the residue R113 to an alanine in the N domains of wild-type p97 and performed a previously established fluorescence-based substrate unfolding assay (Fig. 3a)[16]. In this assay, the decrease in the fluorescence signal corresponds to the unfolding of the polyubiquitinated Ub-Eos substrate (Fig. 3b). As shown in the results, the unfolding activity of R113A was greatly reduced to a similar level as the A232E/E578Q double mutant, suggesting that the interaction is indeed important for the unfolding activity of human p97

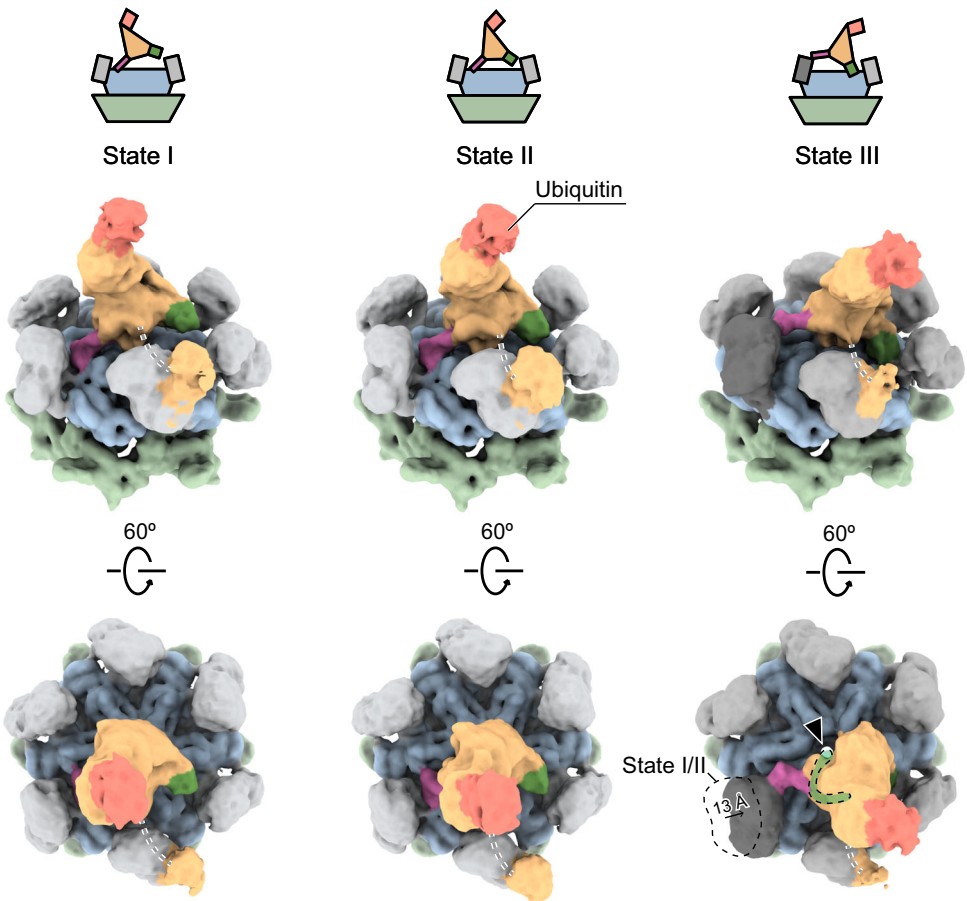

**Fig. 2 Three conformational states of human p97 in complex with Npl4/Ufd1 and polyubiquitinated Ub-Eos.** Single-particle cryo-EM maps (unsharpened, 0.009 threshold) of human p97 in complex with Npl4/Ufd1 and polyubiquitinated Ub-Eos. Three conformational states were resolved using 3D classification. Each state is in a similar conformation to one of the states in the complex of p97 and Npl4/Ufd1 without the substrate (compare to Fig. 1b). One ubiquitin molecule (pink) was observed to bind to the top of Npl4. One of the N domains in State III (dark gray) shifted and interacted with ZF1 (zoomed in Supplementary Fig. 4b). The color scheme used for the other domains is the same as that used in Fig. 1. The position of Npl4 relative to p97 is illustrated in the cartoons. Unresolved linkers between UBXL and ZF1 are represented by dotted lines. The central pore of the D1 ring in State III is highlighted by an arrowhead. The conserved groove of Npl4 that was found to bind an unfolded ubiquitin in the yeast complex structure[14] was marked by a green band in State III.

(Supplementary Fig. 4c). We docked the yeast Npl4-ubiquitin subcomplex from the previously study[14] to the map of State III. Although we could only resolve one ubiquitin moiety binding to Npl4, the position is similar to that of the yeast complex (Supplementary Fig. 4d). The position of the additional proximal ubiquitin in the yeast complex is also consistent with our map showing at a lower threshold (compare Supplementary Fig. 4d, a). In summary, our structures revealed unexpected conformational changes of Npl4 before ATP hydrolysis and translocation, which were not observed in the fungal homolog[13,14]. The unfolding of ubiquitin did not occur in our complex, suggesting that a different state was captured in our study (see discussion). Most interestingly, our structures uncovered an important role of the zinc finger motifs in Npl4, one of which had been recently identified as the binding site of the disulfiram derivative, CuET[12]. Therefore, we conducted further investigations on how CuET might affect the function of human p97.

**Disulfiram derivative CuET inhibits the function of p97 through copper release.** Recently, the metabolic derivative of the anti-alcohol abuse drug disulfiram, CuET (Supplementary Fig. 5a), has been reported to bind to ZF1 of Npl4[12]. Since the zinc finger motifs in Npl4 play critical roles in different

conformational states, we tested whether CuET affects the unfolding activity of p97. We synthesized CuET using diethyl-dithiocarbamate (DTC) and cupric chloride. At room temperature, CuET forms brown-colored crystals and is soluble in dimethyl sulfoxide (DMSO) (Supplementary Fig. 5b). Unexpectedly, we found that only old CuET solution (left at 4 °C for more than 7 days) could inhibit the unfolding activity of p97 (Fig. 3c, green curve); freshly dissolved CuET showed minimal effect (Fig. 3c, orange curve). It is known that disulfide bonds may form in the presence of DMSO[19]. Therefore, we tested if cupric ions were present in the old CuET solution due to the oxidation of the dithiocarbamate group (Supplementary Fig. 5c). Indeed, as we expected, cupric ions were released from the molecule, and oxidative reagents such as hydrogen peroxide can accelerate the reaction, leading to a dramatic color change in the CuET solution (Supplementary Fig. 5d). The reaction depends on the concentration of the oxidant. With a small amount of oxidant, it can last for days (Supplementary Fig. 5d). We further tested if the released cupric ions could actually inhibit the unfolding activity of p97. Indeed, cupric ions inhibited the activity in a concentration-dependent manner (Fig. 3c, e). The release of cupric ions suggested that there is a cycle of disulfiram, DTC, and CuET (Fig. 3f). To further test the existence of a cycle, we introduced additional DTC into the unfolding assay in the presence of old CuET

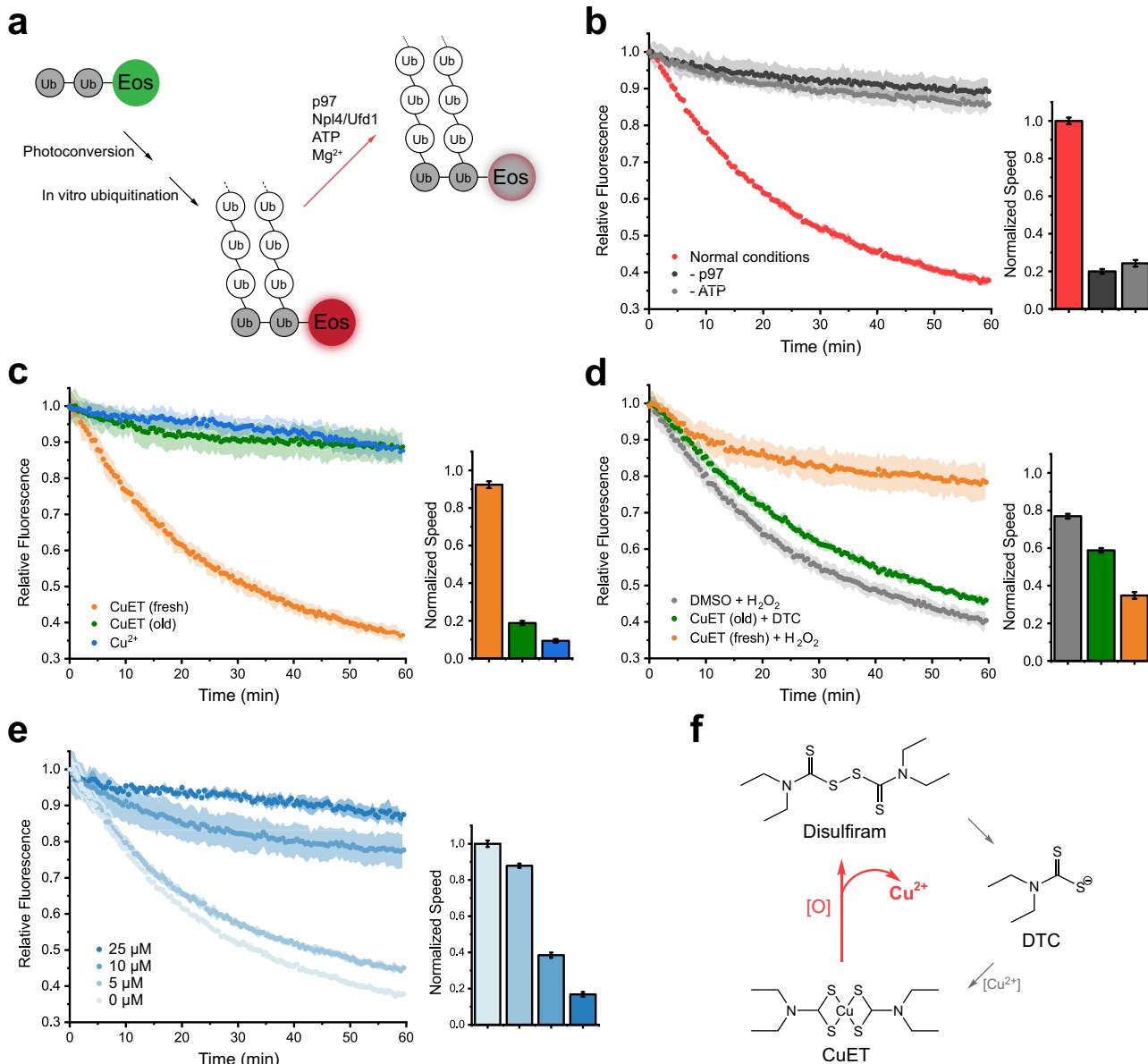

**Fig. 3 Copper released from the disulfiram derivative CuET under oxidative conditions inhibits the unfolding activity of p97. a** A diagram showing the established substrate unfolding assay of p97. The unfolding of ubiquitinated Eos corresponds to a decrease in the fluorescence signal (red arrow). **b** The unfolding activity of wild-type p97 in the presence of Npl4/Ufd1, 5 mM ATP, 20 mM MgCl₂ and ubiquitinated Eos (normal conditions). The relative fluorescence signal (starting at the timepoint zero) was monitored for 60 min. The initial velocity of the reaction was linearly fitted using the data points from the first 10 min and plotted in the bar graph that was normalized against the normal condition. **c** The unfolding activity and initial velocity of p97 in the presence of fresh or old CuET (50 μM) or cupric ions (25 μM). **d** The unfolding activity and initial velocity of wild-type p97 in the presence of fresh or old CuET (50 μM) and an oxidative (H₂O₂, 100 μM) or a reducing (DTC, 100 μM) reagent. **e** The unfolding activity of p97 is inhibited by cupric ions in a concentration-dependent manner. **f** The cycle of disulfiram derivatives. CuET can release cupric ions under oxidative conditions (red arrow). The curves in panels **b–e** are presented as mean values ± SD from triplicate experiments. The error bars of the columns represent the SEM of the linear fitting.

solution. As expected, DTC can rescue the unfolding activity of p97 by chelating the released cupric ions (Fig. 3d, green curve). We also introduced hydrogen peroxide into the unfolding assay in the presence of fresh CuET solution and found that hydrogen peroxide can inhibit the activity by triggering the release of cupric ions (Fig. 3d, orange curve). Furthermore, we measured the unfolding activity of p97 in the presence of other metal ions at the same concentration (25 μM). Interestingly, cupric ions appeared to be the most potent inhibitor in vitro (Supplementary Fig. 5e).

**Cupric ions interact with the zinc finger motifs of Npl4.** It is known that copper can interact with cysteine residues in zinc

finger motifs[20]. Npl4 has three zinc finger motifs: ZF1 and ZF2 bind to the D1 ring and the N domains of p97; NZF binds to the polyubiquitin chains. We therefore tested if cupric ions affected the complex formation of p97, Npl4/Ufd1, and polyubiquitinated Ub-Eos. We formed the complex in the presence of cupric chloride, followed by size-exclusion chromatography and SDS-PAGE. The results showed that cupric ions do not affect the binding between Npl4 and p97 (Supplementary Fig. 6a), nor do they affect the binding between Npl4 and the polyubiquitinated Ub-Eos (Supplementary Fig. 6b). To further quantify the interaction between Npl4 and the copper compounds, we performed a series of isothermal titration calorimetry (ITC) experiments

(Supplementary Fig. 7). The results showed that freshly dissolved CuET does not interact with Npl4 (Supplementary Fig. 7c). Instead, cupric glycinate (used as a replacement for cupric ions) binds to Npl4 at an apparent dissociation constant (Kd$^{app}$) of $2.8 \pm 1.1\ \mu M$ (Supplementary Fig. 7d). By comparison, the binding between cupric glycinate and p97 is much weaker, with a Kd$^{app}$ of $69 \pm 18\ \mu M$ (Supplementary Fig. 7e). We then tested the binding between cupric glycinate and Npl4 bearing zinc finger motif mutants (Supplementary Fig. 7a). With a ZF1 mutation, the binding constant is $7.8 \pm 2.0\ \mu M$; with ZF2 mutation, the binding constant is $8.7 \pm 1.8\ \mu M$; and with both ZF1 and ZF2 mutated, the binding constant is $20 \pm 6.5\ \mu M$ (Supplementary Fig. 7f–h). Taken together, the results suggested that cupric ions interact with Npl4 primarily through ZF1 and ZF2.

To test if the interaction between cupric ions and zinc finger motifs of Npl4 inhibits the unfolding activity of p97, we performed a substrate unfolding assay in the presence of cupric ions and used synthesized zinc finger motifs with D-amino acids (Supplementary Fig. 7a, b) to rescue the activity (Fig. 4a, b). The rationale is that the synthesized D-zinc finger motifs can interact with cupric ions in the same way as the normal L-zinc finger motifs but cannot interact with p97 due to inverted chirality; hence, the seesaw motion of Npl4 will not be disturbed. Therefore, if cupric ions inhibit the unfolding activity of p97 by interacting with the zinc finger motifs of Npl4, the synthesized D-zinc finger motifs should be able to rescue the reactivity. Indeed, both D-ZF1 and D-ZF2 could rescue the unfolding activity of p97, whereas the control D-peptide from Npl4 (Supplementary Fig. 7a, b) could not (Fig. 4a, b).

It has been reported that copper ions could disrupt the zinc finger structure by mediating the oxidation of cysteine residues[20]. Therefore, we further analyzed the reaction between the copper compounds and synthesized zinc finger motifs using high-performance liquid chromatography (HPLC). The results showed that after 60 min of incubation at a 50:1 (Cu$^{2+}$ to peptide) molar ratio, the freshly dissolved CuET did not react with either zinc finger motif (Fig. 4c). By contrast, cupric ions could react with both peptides in a concentration-dependent manner after 15 minutes of incubation, suggesting that the structures of both zinc finger motifs can be disrupted by cupric ions (Fig. 4d). To analyze how the zinc finger motifs in Npl4 react with cupric ions, we performed an iodoacetyl-modified probe-based quantitative mass spectrometry analysis for the cysteine residues in the zinc finger motifs of Npl4. We first incubated purified Npl4 with or without cupric chloride for 30 minutes at room temperature, followed by the following parallel treatment: cysteine labeling, trypsin digestion, and isotope labeling by light and heavy formaldehyde. The reactants were mixed for quantitative liquid chromatography with tandem mass spectrometry (LC-MS/MS) analyses (Fig. 4e). The results showed that the cysteine residues in both zinc finger motifs were much less labeled (light/heavy (L/H) ratio less than 1) after incubation with cupric ions, suggesting that those cysteine residues had been oxidized and were not subjected to the labeling reaction (Fig. 4f).

**CuET treatment increases cellular concentration of cupric ion and decreases cell survival in a redox-dependent manner.** Next, we tested whether CuET treatment can increase the cellular concentration of cupric ion, and how does it respond to cellular redox state. Tert-butyl hydroperoxide (tBHP) was reported to increase the cellular level of reactive oxygen species (ROS)[21], and was further confirmed in our experiment (Supplementary Fig. 8d). Three groups of A549 cells were treated with $5\ \mu M$ CuCl$_2$, $5\ \mu M$ CuET, or $5\ \mu M$ CuET (pretreated with tBHP), respectively. The cellular copper level was analyzed by inductively

coupled plasma mass spectrometry (ICP-MS). The results showed that both CuET and CuET plus tBHP treatments increased cellular copper level significantly as compared to CuCl$_2$ treatment, suggesting that CuET can bypass the copper transporter system on the cell membrane similar to other copper chelators[22] (Fig. 5a). More importantly, with tBHP pretreatment the cellular copper level was significantly higher than that of CuET treatment alone ($n = 3$, $p < 0.05$), suggesting that the oxidative release of copper from CuET also occurs in the cellular environment (Fig. 5a). To further test whether the copper ion has dissociated from the CuET within the cell, we setup four groups of cells with and without CuET and tBHP treatment, respectively. The bioavailable copper in these cells were evaluated by the phosphorylation level of ERK1/2 based on the previous study[22] (Fig. 5b). The results showed that both CuET and CuET plus tBHP treatments increased bioavailable copper level as compared to the cells without CuET treatment. Again, CuET plus tBHP treatment exhibited significant higher phosphorylation level of ERK1/2 than CuET treatment alone ($n = 3$, $p < 0.01$, Fig. 5b, c). In addition, the accumulation of polyubiquitinated (poly-Ub) proteins of these four groups of cells were examined. Higher levels of poly-Ub proteins were observed in both CuET and CuET plus tBHP treated cells, suggesting impaired ubiquitin-proteasome system upon CuET treatment (Supplementary Fig. 8h). Furthermore, we examined the cytotoxicity effect of CuET treatment with and without oxidative stress using both HeLa and A549 cells. Individual dose response curves of CuET and tBHP were determined for HeLa and A549 cells, respectively, to find the optimal concentrations of the treatment (Supplementary Fig. 8a–c). Oxidative stress was induced by treating the cells with tBHP for 6 h. CuET (250 nM) was introduced after the treatment, and the cells were incubated for another 20 h; then, an MTT assay was used to monitor cell survival (Supplementary Fig. 8e). Compared to the control cells that had not been subjected to oxidative stress, the cell survival was significantly lower ($n = 4$, $p < 0.01$) for both CuET-treated HeLa and A549 cells (Supplementary Fig. 8f, g). Our results suggested that CuET is more toxic to the two cancer cell lines under the oxidative condition, likely due to the oxidative release of cupric ion from CuET in the cytoplasm.

**Conformational locking of p97-Npl4/Ufd1 in the presence of cupric ion.** Finally, we determined the single-particle cryo-EM structure of the p97-Npl4/Ufd1 complex in the presence of cupric ions (Fig. 5d). With extensive 3D classifications, we only observed one dominating conformation that is similar to State II of the same complex without cupric ions (Fig. 1b and Supplementary Fig. 9). The final reconstruction reached 3.5 Å, the highest resolution in this study, but the Npl4 part of the cryo-EM map is less well resolved than that of the other regions. Specifically, the ZF2 motif was not resolved in the cryo-EM map, suggesting that under the experimental conditions, ZF2 may have been disrupted by cupric ions and could not bind to the D1 ring of p97. Therefore, we did not obtain a conformation similar to State III of the same complex without cupric ions. Since the seesaw motion of Npl4 is important for the unfolding activity of human p97, the lack of other states suggested a conformational lock, leading to the inhibition of activity.

## Discussion

In this study, we focused on structural and functional analyses of human p97 and its cofactor Npl4/Ufd1. We discovered three major conformational states of the p97-Npl4/Ufd1 complex both in the presence and absence of the polyubiquitinated substrate and ATPγS. Our structures of human p97 complexes suggest a seesaw motion of Npl4 on top of the D1 ring. We further showed

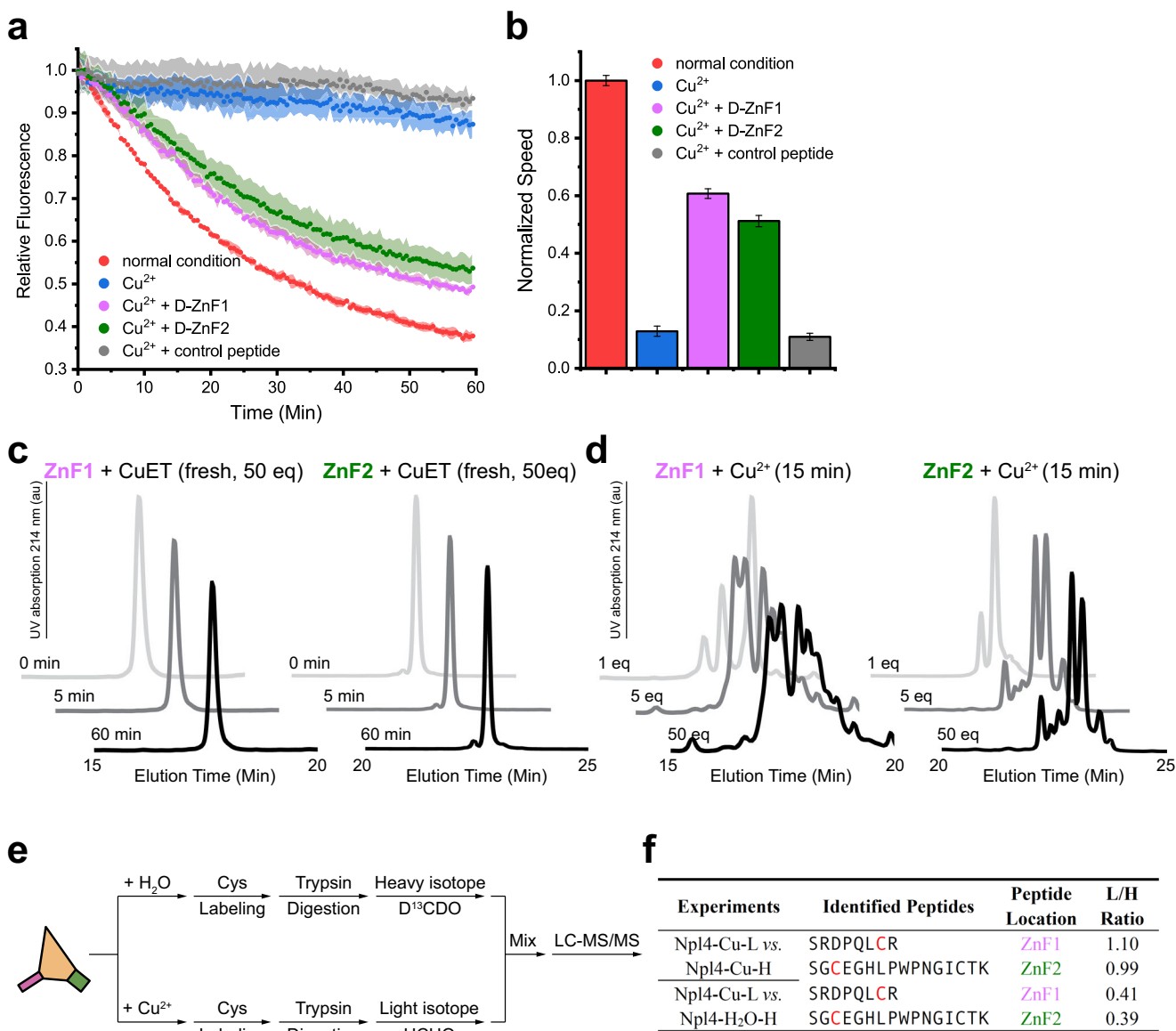

**Fig. 4 Copper interacts with the zinc finger motifs of Npl4. a** Zinc finger motifs of Npl4 rescue the unfolding activity of p97 in the presence of cupric ions. **b** The initial velocity corresponds to panel **a**. The curves are presented as mean values ± SD from triplicate experiments. The error bars of the columns represent the SEM of the linear fitting. **c** HPLC profiles of synthesized ZF1 and ZF2 peptides mixed with fresh CuET (at 1:50 molar ratio) over 60 min. **d** HPLC profiles of synthesized ZF1 and ZF2 peptides mixed with cupric chloride at different ratios for 15 min. **e** A diagram showing the experimental steps of quantitative mass spectrometry using light and heavy formaldehyde to probe the interactions between cupric ions and the cysteine residues in the zinc finger motifs of Npl4. Cupric ions oxidize the cysteine residues of the zinc finger motifs and decrease the labeling percentage. **f** Results from the quantitative mass spectrometry (panel **e**). Labeled cysteine residues are highlighted in red. The first experiment was used as a control, in which both light and heavy labeling were treated with cupric ions, and the expected L/H ratio was 1.

that the seesaw motion is essential for unfolding the substrate. Note that one of the states exposes the central pore of the D1 ring. This dynamic movement leads to a model: the seesaw motion of Npl4 may facilitate the initial unfolding of ubiquitin by exposing the pore of the D1 ring. The partially unfolded or destabilized ubiquitin is then captured by Npl4 or directly inserted into the pore, which may or may not lead to a structure similar to that observed for yeast Cdc48. The latter possibility suggests a different mechanism between yeast and humans, which cannot be ruled out at this stage. There are still missing pieces in this model, and further structural studies are required to elucidate the details.

We also found that at least one zinc finger motif interacts with the N domains of p97 (Supplementary Fig. 4b). It is known that N domains change conformations upon ATP hydrolysis[15,23].

Therefore, the seesaw motion we observed here may be coupled with ATP hydrolysis that occurs in the D1 and D2 rings through the interactions between the zinc finger motifs and the N domains. In such a case, ATP hydrolysis would be the driving force for the seesaw motion of Npl4. What we captured without triggering ATP hydrolysis are three intermediate conformations of the motion. Note that this does not contradict the model of translocation initiation that is proposed above, as initiation may require ATP hydrolysis as well.

We showed that CuET could disrupt the seesaw motion of Npl4 through the release of copper ions under oxidative conditions (Fig. 5e). This is likely a mechanism linking disulfiram to p97 inhibition. Disulfiram has long been known to have anticancer activity[24]. Recently, the metabolic derivative of disulfiram,

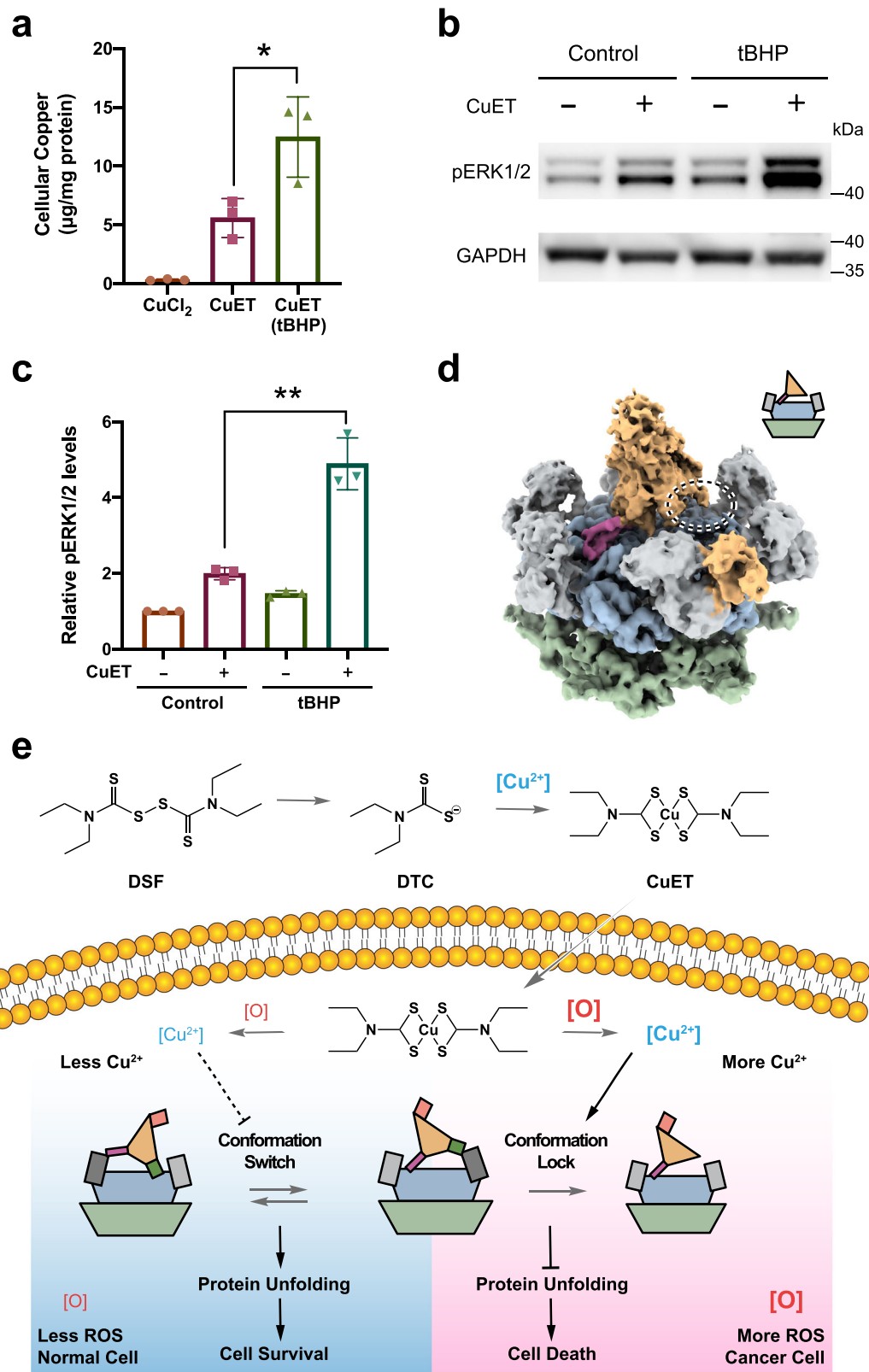

CuET, has been found to target cancer though Npl4[12,25]. Our biochemical and structural data suggested that cupric ions, not CuET, interact with the zinc finger motifs of Npl4; as a result, cupric ions inhibit the unfolding activity of p97 through a conformational lock. We discovered that cupric ions are released by CuET under oxidative conditions. The velocity of the reaction depends on the concentration of the oxidant and can last for days

(Supplementary Fig. 5d). Such conditions may exist in cancer cells and tissues that are often under oxidative stress[26]. Therefore, we propose that instead of directly targeting Npl4, CuET acts as a shuttle for cupric ions to cross the cell membrane, similar to the mechanism of various copper complexes of bis(thiosemicarbazones) used for therapeutics and imaging as reported before[27–29]. Furthermore, we showed that once in the cytosol, CuET releases

**Fig. 5 Conformational lock induced by copper released from CuET. a** Quantification of cellular copper level using ICP-MS. A549 cells were treated with 5 μM CuCl₂, 5 μM CuET, or 5 μM CuET (pretreated with 2.5 mM tBHP), respectively (n = 3 biological replicates, mean ± SD, *p = 0.037, two-sided Student's t-test). **b** Quantification of cellular bioavailable copper using western blot against ERK1/2[22]. A549 cells were divided into four groups, pretreated with and without 2.5 mM tBHP, and treated with and without 5 μM CuET, respectively. **c** Quantification of the Western blot in panel **b** using ImageJ[62] (n = 3 biological replicates, mean ± SD, **p = 0.002, two-sided Student's t-test). **d** Single-particle cryo-EM map (unsharpened, 0.00715 threshold) of human p97 in complex with Npl4/Ufd1 in the presence of cupric ions (100 μM). The color scheme is the same as that used in Fig. 1. In contrast to the complex without cupric ions, only one conformational state was resolved. Note that ZF2 was not resolved in the cryo-EM map (the circle highlights this area). **e** A summary of the findings in this work. CuET bypasses the copper transporter system and releases cupric ions under oxidative conditions such as in cancer cells, inhibiting the unfolding activity of p97 through conformational locking.

cupric ions depending on the cellular oxidative status, which then impair the function of essential cellular machines such as p97. Although there are many cellular processes that may be affected by cupric ions[30,31], we speculate that p97 may be one of the major targets, as it is very abundant and plays a key role in cellular proteostasis.

It is noteworthy that most previous studies reported that cupric ions can be released from the copper complexes of bis(thiosemicarbazones) under hypoxia conditions[32,33]. Oxidative release has only been reported in vitro[34]. Nevertheless, it is clear that those compounds including CuET can bypass the copper transporter system and penetrate the cell membrane despite the strict regulation of copper transport[35]. It has been reported that the intracellular copper concentration increased rapidly when the cells were treated with disulfiram in medium containing cupric chloride, and the intracellular copper uptake could be blocked by coincubation with bathocuproinedisulfonic acid, a nonmembrane-permeable copper chelator[36]. Such a copper shuttling mechanism has been reported for radiopharmaceuticals used in both PET imaging and the treatment of hypoxic tumors[33,37]. In addition, copper-based drugs have been developed for the treatment of other diseases and cancers[38]. It is particularly interesting to explore the relationship between these compounds and p97 in the following-up studies. Moreover, mutations of p97 has been linked to neurodegenerative diseases such as familial amyotrophic lateral sclerosis[7]. The copper chelator known as diacetyl-bis(N(4)- ethylthiosemicarbazonato) copper (CuATSM), has been reported recently to be efficacious in multiple transgenic SOD1 models of amyotrophic lateral sclerosis in multiple studies[39–45]. Similar compounds have also been shown to decrease the abundance of toxic oligomers in the brains of transgenic Alzheimer's disease model mice[46]. It will be interesting to test whether disulfiram or CuET has a similar effect in the disease models. Further investigations are required to elucidate the relationship between cytoplasmic cupric ions and the function of p97. In summary, our findings suggested a mechanism of disulfiram's broad anticancer activity when used in combination with copper compounds, which may be of great clinical interests.

## Methods

**Protein expression and purification.** Full-length human p97 including mutant A232E/E578Q, Ufd1, Npl4, as well as wild-type and mutant Npl4 fragments (residues 129–580) were overexpressed in *E. coli* BL21-DE3-RIPL cells. p97 was cloned into pET-47b vector with an N-terminal cleavable His-tag. All mutants of p97 were generated by site-directed mutagenesis using wild-type p97 as a template (Supplementary Table 3). Two plasmids containing full-length untagged Npl4 and C-terminal His-tagged Ufd1 were purchased from Addgene. Npl4 fragments were expressed in pET-28a vector with an N-terminal His-SUMO tag. Purification protocols were similar to those published before[47]. Briefly, *E. coli* cells were grown in LB with auto-induction supplement for 16 h at 30 °C. Cells were pelleted at 5000 × g and resuspended in Lysis Buffer (50 mM Tris, pH 8.0, 300 mM KCl, 20 mM imidazole, 1 mM DTT, and 1 mM MgCl₂). To purify the Npl4/Ufd1 complex, the resuspended cells of respective construct were mixed at this step. 1 mM PMSF was added to the suspensions, and cells were lysed by sonication. Lysates were cleared by centrifugation for 30 min at 30,000 × g. then the supernatants were flowed through a Ni-NTA gravity column twice at 4 °C. Beads were washed with Wash Buffer (50 mM Tris, pH 8.0, 150 mM NaCl, 20 mM imidazole

and 1 mM MgCl₂). The His-tagged proteins were eluted in Elution Buffer (50 mM Tris, pH 8.0, 300 mM NaCl, and 400 mM imidazole). For p97 and Npl4 fragments, HRV3C protease and SUMO protease (Ulp1p) were added to the eluate, respectively, and dialyzed in SEC Buffer (25 mM Tris, pH 8.0, 150 mM NaCl, 1 mM MgCl₂, and 0.5 mM Tris(2-carboxyethyl)phosphine (TCEP)). Npl4/Ufd1 complex were directly dialyzed in the SEC Buffer. Npl4/Ufd1 complex and Npl4 fragments were further purified by a Superdex 200 size-exclusion column (GE Healthcare) equilibrated in the SEC Buffer. p97 was purified by a Superose 6 column (GE Healthcare) equilibrated in the SEC Buffer.

**Preparation of polyubiquitinated Ub-Eos.** Ub-Eos was constructed by fusing two tandem ubiquitin to the N-terminus of fluorescent protein mEos3.2. The polyubiquitinated Ub-Eos was prepared similar to previously described[16]. Briefly, ubiquitination reaction was carried out with final concentrations of 20 μM Ub-Eos, 1 μM E1, 20 μM gp78RING-Ube2g2, and 500 μM ubiquitin in 20 mM Hepes, pH 7.4, 150 mM KCl, 10 mM ATP, and 10 mM MgCl₂ at 37 °C overnight. To purify ubiquitinated Eos3.2 from free ubiquitin chains, the reaction mixture was incubated with Ni-NTA resin, eluted with 300 mM imidazole, and run over a Superdex 200 size-exclusion column equilibrated in the SEC Buffer. Fractions with long ubiquitin chain (over 10 Ubs) were pooled and concentrated with centrifugal filter units, followed by flash frozen.

**p97 complex formation.** To form p97-Npl4/Ufd1 complex, Npl4/Ufd1 was added at a three folds molar excess to p97 hexamer, and proteins were incubated with 1 mM ATPγS for 60 min before gel filtration. After gel filtration, all proteins were flash frozen in liquid nitrogen. For p97-Npl4/Ufd1-Ub-Eos complex, a mutant p97 bearing A232E and E578Q mutations was used instead of wild-type p97. To form the ternary complex, p97 (A232E/ E578Q)-Npl4/Ufd1 complex was first prepared as described above, then polyubiquitinated Ub-Eos was added at a two-fold molar excess to p97 (A232E/E578Q)-Npl4/Ufd1 complex. After gel filtration, all proteins were flash frozen in liquid nitrogen.

**Specimen preparation for single-particle cryo-EM.** For p97-Npl4/Ufd1 complex, before preparing grids for cryo-EM, the complex was concentrated to ~20 mg/mL, and incubated with 5 mM ATPγS (90% pure, Sigma) for 30 min at room temperature. For p97 (A232E/E578Q)-Npl4/Ufd1-Ub-Eos complex, the sample was concentrated to 20 mg/mL without adding additional nucleotide. For p97-Npl4/Ufd1 complex in the presence of cupric ion, the complex was concentrated to ~20 mg/mL, and incubated with 5 mM ATPγS (90% pure, Sigma) and 100 μM cupric ion for 30 min at room temperature. To relief the preferred orientations, IGEPAL CA-630 (Sigma) was added to the samples to a final concentration of 0.05% immediately before grid freezing. The freezing was performed using a Vitrobot mark IV (Thermo Fisher) operating at 8 °C and 100% humidity. Samples (3.5 μL) were applied to a non-glow-discharged Quantifoil Au 1.2/1.3 grid. The grid was blotted for 1 s using standard Vitrobot filter paper (Ted Pella, 47000-100) and then plunge frozen in liquid ethane.

**Data collection for single-particle cryo-EM.** Optimized frozen grids were shipped to National Cryo-Electron Microscopy Facility for data collection. All datasets were acquired as movie stacks with a Titan Krios electron microscope operating at 300 kV, equipped with either a Gatan K2 Summit or K3 direct detection camera. A single stack typically consists of 40 frames with a total exposure around 50 electrons/Å². The defocus range was set at −1.0 to −2.5 μm. See Supplementary Table 1 for the details.

**Image processing.** Movie stacks were subjected to beam-induced motion correction using MotionCor2[48]. CTF parameters for each micrograph were determined by CTFFIND4[49]. The following particle picking, two- and three-dimensional classifications, and three-dimensional refinement were performed in RELION-3[50]. Briefly, particle picking was done by manually choosing ~2000 particles and generating templates through 2D classification for the following automatic picking. False-positive particles or particles classified in poorly defined classes were discarded after 2D classification. The initial 3D classification was performed on a binned dataset with the previously reported p97 structures as the

reference model[15]. The detailed data processing flows are shown in Supplementary Figs. 1, 3, and 9. To make sure that the 3D classification did not miss any major conformations, additional runs were performed asking for different number of classes and using different regularization parameters (T ranges from 2 to 6), which all ended up with the same number of major classes. Since we did not observe a dependent on the regularization parameter, the results from the default value (T = 4) were shown in the figures. To improve the resolution of Npl4/Ufd1 part of the map, masked refinement, focused classification, and multibody refinement in RELION were performed, but the resulting maps were not improved. Data processing statistics are summarized in Supplementary Table 1. Reported resolutions are based on Fourier shell correlation (FSC) using the FSC = 0.143 criterion. Local resolution was determined using ResMap[51] with half-reconstructions as input maps. Histograms and directional FSC plots for the cryo-EM maps generated using the online 3DFSC server (https://3dfsc.salk.edu/upload/)[52] were summarized in Supplementary Fig. 10.

**Model building, refinement, and validation**. Model building was based on the existing crystal structures of human p97[5] (PDB code: 3CF3). A homology model of human Npl4 was built using SWISS-MODEL[53] based on the crystal structure of thermophilic fungi Npl4[13] (PDB code: 6CDD). The existing p97 model and the homology model were first docked into the cryo-EM maps as rigid bodies using UCSF Chimera[54]. The p97 part was then manually adjusted residue-by-residue to fit the density using COOT[55], and was subjected to global refinement and minimization in real space using the real space refine module in Phenix[56]. The Npl4 part was kept as a rigid body during the process. The statistics of model refinement and geometry (p97 part) is shown in Supplementary Table 2.

**Cys reactivity validation through iodoacetyl-modified probe-based LC-MS/MS**. The quantitative chemical proteomics was performed as recently described[57]. 200 μL Npl4 (0.11 mg/mL in 25 mM Hepes, pH 7.6, 150 mM NaCl,) was directly incubated with 1 μL CuCl₂ (2 mM) or 1 μL ddH₂O for 30 min at 25 °C, respectively. After the treatment with or without copper, iodoacetamide (IA probe) was added to a final concentration of 100 μM, and incubated for 60 min at 25 °C. Then, 10 μL DTT (200 mM) was added to each sample at 37 °C for 30 min to quench the reaction. Each solution was supplemented with 8 M Urea and then diluted with 1 mL 100 mM triethylammonium bicarbonate buffer (TEAB) to a final urea concentration of 2 M. 10 μL mass spectrometry grade trypsin (0.5 μg/μL, ratio of samples: trypsin = 1: 40 (w/w), Promega, V5280) was used to digest protein samples overnight at 37 °C. 8 μL of 4% (v/v) 'light' (Sigma, F1635) or 'heavy' formaldehyde-13C, d2 (Sigma, 596388) was added to the reaction with or without the copper treatment, respectively. 8 μL of sodium cyanoborohydride (0.6 M) were added and the reaction was incubated at 25 °C for 1 h before quenching with 32 μL of 1% ammonia followed by 16 μL of 5% formic acid. The corresponding 'light' and 'heavy' sample were combined and centrifuged (1400 × g, 2 min). Mixed peptides were dried and stored at −20 °C until LC-MS/MS analysis.

LC-MS/MS data were analyzed by ProLuCID[58] with static modification of cysteine (+57.0215 Da) and variable oxidation of methionine (+15.9949 Da). The isotopic modifications (+28.0313 and +34.0631 Da for light and heavy labeling respectively) were set as static modifications on the N-terminal of a peptide and lysine residues. Additional 357.17223 Da of IA probe was set as variable modifications on cysteines. The searching results were filtered by DTASelect[59] and peptides were also restricted to fully tryptic with a defined peptide false-positive rate of 1%. The ratios (L/H) of reductive dimethylation were quantified by the CIMAGE software as described before[60].

**Substrate unfolding assay**. The polyubiquitinated, photo-converted Ub-Eos was prepared as described[17]. Experiments were carried out in Reaction Buffer (20 mM Hepes, pH 7.4, 150 mM KCl, 20 mM MgCl2, 1 mg/mL BSA) supplemented with an ATP regeneration mixture (5 mM ATP, 30 mM creatine phosphate, and 50 μg/mL creatine phosphokinase). Proteins were pre-incubated in a 96-well plate (Fisherbrand FB012931) for 10 min at 37 °C before adding the ATP regeneration mixture to initiate the reaction. Final concentrations of the reactants were 20 nM substrate, 400 nM p97 in hexamer, and 300 nM Npl4/Ufd1. Fluorescence signal was monitored using a TECAN safire2 plate reader at 540 nm excitation and 580 nm emission wavelength and 30 s intervals for 60 min. Each reaction was repeated three times. Background fluorescence was measured by mixing the same amount of substrate with 6 M guanidine-HCl and was subtracted from the average of the experimental groups. Normalized fluorescence and the initial velocity of the reaction (first 20 data points corresponding to 10 min of reaction) were plotted and fitted using OriginPro (OriginLab).

**Isothermal titration calorimetry**. Isothermal titration calorimetry measurements were carried out on an ITC200 Microcalorimeter (GE Healthcare). p97 and Npl4 fragments were dialyzed in a buffer containing 25 mM HEPES (pH 7.4) and 150 mM NaCl. Sufficient amount of 1 mM Cu(Gly)₂ in the injection syringe was titrated to the sample cell containing 0.02 mM of p97 or Npl4 fragments to achieve a complete binding isotherm. All binding experiments were performed at constant temperature of 25 °C. A total of 20 injections of 2.0 μL were dispensed with a 5-second addition time and a spacing of 120 s. All experiments were repeated at least three times. Data were analyzed and the titration curves were fitted using MicroCal Origin software assuming a single binding site mode.

**Peptide synthesis**. All peptides used in this work were synthesized using standard Fmoc SPPS protocols under microwave conditions (CEM Liberty Blue)[61]. Rink Amide AM resin was first swelled in DMF for 10 min. Every coupling cycle was executed programmatically. In general, the deprotected condition is 10% piperidine in DMF with 0.1 M Oxyma (1 min at 90 °C) and the amino acid coupled condition is 4 folds of 0.2 M Fmoc-protected amino acid, 1.0 M DIC, and 1.0 M Oxyma in DMF (10 min at 50 °C for His and Cys, 90 °C for other residues). After the completion of SPPS, the peptide-resin was transferred into customized sand core funnel and treated with the cleavage cocktail (TFA/H₂O/thioanisole/EDT 87.5/5/5/2.5, v/v/v/v) for 2 h at room temperature. Crude peptides were precipitated with cold diethyl ether and dissolved in water (containing 0.1% TFA) mixed with acetonitrile (containing 0.1% TFA) and purified by semi-preparative RP-HPLC.

**Measurement of tBHP induced oxidatively modified proteins**. Two experimental groups were designed, namely the tBHP group and the Control group. A549 cells in the tBHP group were pretreated with 2.5 mM tBHP (Sigma, 458139) for 2 h, and the control group was treated with ddH₂O instead. Cells were then washed twice with ice-cold PBS and harvested with the addition of 100 μL NP-40 lysis buffer (50 mM Tris-HCl, pH 8.0, 150 mM NaCl, 1.0% NP-40, 0.1% Triton X-100, and cocktail protease inhibitors). Cell lysates were analyzed for oxidatively modified proteins using the Oxyblot Protein Oxidation Detection Kit (Millipore, S7150). Protein concentrations were measured using BCA assay (BioRad) to ensure that equal amount of samples were subjected to the Oxyblot kit.

**Measurement of intracellular copper level**. Three experimental groups were designed, namely CuCl₂ group, CuET group and CuET (tBHP) group. Cells in the CuET (tBHP) group were pretreated with 2.5 mM tBHP (Sigma, 458139) for 2 h, which was reported to increase the intracellular ROS level by 40%[21]. The other two groups of cells were pretreated with ddH₂O. After pretreatment, the three groups of cells were treated with 5 μM CuCl₂, 5 μM CuET and 5 μM CuET for 3 h, respectively. Then, cells were harvested using trypsin-EDTA solution, and washed twice with metal-free Hanks' balanced salt solution (HBSS). Next, 300 μL lysis buffer (Beyotime, P0013) was added to each group of cell pellets. Aliquots of the three cell suspensions were collected to measure the protein content. Finally, cell suspensions were digested with 5 mL concentrated nitric acid and then analyzed for copper content using ICP-MS.

**Measurement of intracellular bioavailable copper**. The intracellular bioavailable copper content was evaluated by the phosphorylated level of ERK1/2 according to the previous study[22]. Four experimental groups were designed, namely DMSO group, CuET group, tBHP group and CuET (tBHP) group. First, tBHP group and CuET (tBHP) group were pretreated with 2.5 mM tBHP (Sigma, 458139) for 2 h, and the other two groups were treated with ddH₂O. After pretreatment, the four groups were treated with DMSO, 5 μM CuET, DMSO and 5 μM CuET for 3 h, respectively. Then the cells were washed twice with PBS and harvested with the addition of 100 μL phosphosafe extraction reagent buffer (Merck, lot: 3388). After centrifugation, the cell lysates were mixed with 4× loading buffer and heated at 95 °C for 10 min. The phosphorylation level of ERK1/2 were then detected using Western blot (pERK1/2 antibody, Cell Signaling, 9101 S, 1:1000 dilution) and evaluated using ImageJ[62].

**MTT Assay for CuET and copper toxicity**. The MTT assay to evaluate CuET toxicity under oxidative conditions in cancer cells was modified from previous protocols[63]. Briefly, cells were plated on 96-well tissue culture plates (Fisher, FB012931) at 5000 cell per well and grown at 5% CO₂ at 37 °C. The next day, oxidative stress was achieved by treating the cells with growth medium containing 0, 25, or 50 μM TBH70X (Sigma, 458139). After 6 h, the TBH70X-containing medium was removed and replaced with medium containing 0, 0.1 or 0.25 μM CuET. The DMSO control contained 100 μL culture medium with 0.05 μL DMSO (Fisher, BP231100). After 18 h of CuET treatment, 20 μL of 5 mg/mL MTT (Sigma, M2128) was added to each well and the cells were further incubated for 3.5 h in 37 °C incubator. To dissolve the formazan crystals, the culture medium was carefully removed and 150 μL of MTT solvent (4 mM HCl, 0.1% NP-40 in isopropanol) was added to each well with 15 min shaking at room temperature. The absorbance was measured at 590 nm with a reference wavelength of 620 nm with a SAFIRE II plate reader (Tecan, Männedorf, Switzerland). Each experimental group contained four biological replicates with the error bar indicating standard deviation. The cell survival rate was calculated by the following equation: cell survival rate (%) = (absorbance of experimental group/absorbance of control group) × 100%.

**Reporting summary**. Further information on research design is available in the Nature Research Reporting Summary linked to this article.

## Data availability

cryo-EM maps have been deposited in the Electron Microscopy Data Bank (EMDB) under the accession codes EMD-21824 (p97-Npl4/Ufd1, State I), EMD-21825 (p97-Npl4/Ufd1, State II), EMD-21826 (p97-Npl4/Ufd1, State III), EMD-21827 (p97-Npl4/Ufd1-Ub-Eos, State I), EMD-21828 (p97-Npl4/Ufd1-Ub-Eos, State II), EMD-21829 (p97-Npl4/Ufd1-Ub-Eos, State III), EMD-21830 (p97-Npl4/Ufd1 in the presence of cupric ion), and EMD-22521 (p97-Npl4/Ufd1, masked around p97 and applied C6 symmetry). The atomic model of p97 bound to ATPγS has been deposited in the Protein Data Bank (PDB) under the accession code 7JY5. Other data are available from the corresponding authors upon reasonable request. Source data are provided with this paper.

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

## Acknowledgements

Funding for this work was, in part, provided by the Catalyst Award from the Chicago Biomedical Consortium. This work was supported by Chicago Biomedical Consortium Catalyst Award C-086 to M.Z. We thank the National Key R&D Program of China (No. 2017YFA0505200), NSFC (91753205) for financial support. We thank staff at the National Cryo-EM Facility and Min Su at the University of Michigan for the help in cryo-EM data collection. This research was, in part, supported by the National Cancer Institute's National Cryo-EM Facility at the Frederick National Laboratory for Cancer Research under contract HSSN261200800001E.

## Author contributions

M.P., L.L., and M.Z. designed all the experiments and interpreted the results. M.P., Q.Z., Y.Y., and H.A. cloned, expressed and purified all the complexes and carried out related biochemical characterizations including the substrate unfolding assay and the ITC experiments. M.P. and M.Z. performed cryo-EM data collection and processing. Y.Y. synthesized the disulfiram derivative CuET. Y.X. performed the cell survival assay. Q.Z., X.Z., and C.W. performed the iodoacetyl-modified probe-based LC-MS/MS experiments. M.Z., M.P., and Q.Z. wrote the paper. M.Z. and L.L. supervised the project.

## Competing interests

The authors declare no competing interests.
