## [Peer Review File · Nature Communications]

This manuscript has been previously reviewed at another journal that is not operating a transparent peer review scheme. This document only contains reviewer comments and rebuttal letters for versions considered at *Nature Communications*.

Reviewers' Comments:

Reviewer #1:

Remarks to the Author:

I am generally satisfied with the authors' responses, although they did not make any attempt to assess the influence of FSC inflation due to preferred views. Two of the reviewers commented on the lack of an Euler distribution plot in the supplementary figures, and yet the revision does not contain any such plots. Thus, it is difficult to assess possible impact of preferred views on the reported resolutions.

Reviewer #2:

Remarks to the Author:

The authors address all my concerns/comments as well as those of other reviewers.

Congratulations on the manuscript, this is a nice addition to the field. -Ed

Reviewer #4:

Remarks to the Author:

Man Pan, Qingyun Zheng, Yuanyuan Yu and colleagues have presented a good revision of their manuscript on the interaction between Npl4 and p97, and the inhibitory action of the copper derivative of disulfiram, CuET. Their revisions and responses have addressed many of my original comments very nicely. The new data presented in Figure 5a-c is very good. However, I believe there are a few matters which were not addressed sufficiently in the revision. These do not represent major revisions, but I believe they are important:

1. Whilst the use of tBHP to induce oxidative stress may be a common method, data to confirm that it has induced oxidative stress in the authors' cell culture experiments should be included.
2. Broader discussion (with more appropriate citation of existing literature) is needed to address the abundance of literature on copper-containing compounds for treating (and imaging) cancer. Although it is an old review, PMID 21409228 would be a good place to start. Work by authors such as Blower, Dilworth, Ikawa and Fujibayashi needs to be cited.
3. If the use of copper-containing compounds for the treatment of neurodegenerative disease is to be included there are more pertinent papers to cite than reference #29. For example, PMID 24899723 which actually shows that CuATSM delivers bioavailable copper in vivo and PMID 19122148 which relates to an analogous compound and Alzheimer's disease.

Thank you very much for your kind consideration of our manuscript. We have carefully made changes according to the suggestions of the reviewers. We have also updated Figures S1, S3, S8, S9, and make a new Figure S10 to address reviewers' concerns. Here are the detailed responses to the reviewers' comments:

Reviewer #1 (Remarks to the Author):

I am generally satisfied with the authors' responses, although they did not make any attempt to assess the influence of FSC inflation due to preferred views. Two of the reviewers commented on the lack of an Euler distribution plot in the supplementary figures, and yet the revision does not contain any such plots. Thus, it is difficult to assess possible impact of preferred views on the reported resolutions.

We have include the distributions of Euler angles in Figures S1, S3, and S9. In addition, we made a new Figure S10 to show the results of 3D FSC using the online 3DFSC server. The sphericity of all the reconstructions are above 0.9, so there is minimal preferred orientation problem in our samples.

Reviewer #4 (Remarks to the Author):

Man Pan, Qingyun Zheng, Yuanyuan Yu and colleagues have presented a good revision of their manuscript on the interaction between Npl4 and p97, and the inhibitory action of the copper derivative of disulfiram, CuET. Their revisions and responses have addressed many of my original comments very nicely. The new data presented in Figure 5a-c is very good. However, I believe there are a few matters which were not addressed sufficiently in the revision. These do not represent major revisions, but I believe they are important:

1. Whilst the use of tBHP to induce oxidative stress may be a common method, data to confirm that it has induced oxidative stress in the authors' cell culture experiments should be included.

We have include the data in Fig. S8d using the Oxyblot Protein Oxidation Detection Kit (Millipore, S7150).

2. Broader discussion (with more appropriate citation of existing literature) is needed to address the abundance of literature on copper-containing compounds for treating (and imaging) cancer. Although it is an old review, PMID 21409228 would be a good place to start. Work by authors such as Blower, Dilworth, Ikawa and Fujibayashi needs to be cited.

We have included a broader discussion of the copper compounds and cited the papers.

3. If the use of copper-containing compounds for the treatment of neurodegenerative disease is to be included there are more pertinent papers to cite than reference #29. For example, PMID 24899723 which actually shows that CuATSM delivers bioavailable copper in vivo and PMID 19122148 which relates to an analogous compound and Alzheimer's disease.

It is an interesting point since p97 is also involved in neurodegenerative diseases. We have mentioned the treatment of neurodegenerative disease using copper compounds at the end of the discussion to call for follow-up studies.

We thank the reviewers for the valuable comments and suggestions that help improve the manuscript.